# Bioaccessibility and Antioxidant Activity of Polyphenols from Pigmented Barley and Wheat

**DOI:** 10.3390/foods11223697

**Published:** 2022-11-18

**Authors:** Borkwei Ed Nignpense, Sajid Latif, Nidhish Francis, Christopher Blanchard, Abishek Bommannan Santhakumar

**Affiliations:** 1School of Dentistry and Medical Sciences, Charles Sturt University, Locked Bag 588, Wagga Wagga, NSW 2678, Australia; 2National Life Sciences Research Hub, Faculty of Science and Health, Charles Sturt University, Wagga Wagga, NSW 2795, Australia; 3School of Agricultural, Environment and Veterinary Sciences, Charles Sturt University, Wagga Wagga, NSW 2650, Australia; 4Gulbali Institute of Agriculture, Water and the Environment, Charles Sturt University, Wagga Wagga, NSW 2650, Australia

**Keywords:** polyphenols, pigmented cereals, antioxidant, gastrointestinal digestion, bioaccessibility

## Abstract

Polyphenols in pigmented cereals are believed to enhance health outcomes through their antioxidant properties. This study aimed to characterise polyphenols from *Hordeum vulgare* (purple barley), *Triticum turgidum* (purple wheat) and *Triticum aestivum* (blue wheat) in order to evaluate their bioaccessibility and antioxidant activity. An ultra-high performance liquid chromatography mass spectrometry coupled with an online 2,2′-azino-bis (3-ethylbenzothiazoline-6-sulfonic acid) system was used to identify the polyphenols and quantify their relative antioxidant levels. Simulated gastrointestinal digestion of the cereals allowed for the assessment of polyphenol bioaccessibility using benchtop assays. Between cereals, the bioaccessible phenolic content was similar following digestion, but the antioxidant activity was significantly different (purple barley > purple wheat > blue wheat; *p* < 0.01). Among the polyphenols identified, flavan-3-ols and anthocyanins were the least bioaccessible whereas flavones were the most bioaccessible after digestion. This study demonstrated that these pigmented cereal varieties are sources of bioaccessible polyphenols with antioxidant activity. These findings may aid in utilising these pigmented grains for the future design and development of novel functional food products with enhanced health properties.

## 1. Introduction

Polyphenols are plant-derived secondary metabolites that have been shown to exhibit therapeutic and disease-preventive potential through in vitro, ex vivo and in vivo studies [1]. Polyphenols are a heterogenous of group of compounds consisting of phenolic acids which have one aromatic ring, and flavonoids which have at least three rings (A, B and C aromatic rings) [1]. The hydroxyl groups and other functional groups attached to the rings allow the compounds to exert antioxidant activity which in turn reduces oxidative stress, a precursor involved in progression of chronic disease states [2]. Pigmented cereals are gaining attention due to their rich phenolic content and associated bioactivity [3,4]. Cereals displaying black, purple, blue or red pigments have been shown to contain polyphenols, particularly anthocyanins, with potent antioxidant activities including metal ion chelation and free radical scavenging activity [5,6,7].

Pigmented barley (*Hordeum vulgare*) and wheat (*Triticum aestivum* and *Triticum turgidum*) are gaining attention due to their rich phenolic content, health implications and various food applications [8,9,10,11,12,13]. *Hordeum vulgare* is a common hull-less barley that is used in making beverages and other cereal based products [8]. It has been shown that the anthocyanin-rich extracts of hull-less purple barley to exhibit antioxidant and antihypertensive activities [9]. It has also been reported that sourdoughs made from hull-less barley and pigmented wheat flours exhibit antioxidant activity on murine macrophages [10]. Although, nonpigmented barley was used in the study, it can be speculated that the addition of pigmented barley would have provided additional health benefits due to the anthocyanins present. Likewise, pigmented varieties of the common bread wheat (*Triticum aestivum*) offer great nutritional value due to phenolic content [8]. The blue coloured variety is pigmented in the aleurone layer and has been the subject of studies focused on the antioxidant activity of pigmented wheats and selective breeding of new varieties [11,12]. On the other hand, *Triticum turgidum* is a durum wheat originating from Ethiopia [12]. It is the wheat commonly used in making pasta. The purple-coloured durum wheat is rich in phenolic compounds contributing to its antioxidant capacity [13]. However, despite the growing knowledge of the health benefits and food applications there is limited understanding of the bioavailability of phenolic compounds from these three pigmented grains.

In addition to anthocyanins, a variety of polyphenols have been identified in cereal grains, but further investigations are warranted to characterise individual polyphenols present and their levels of bioavailability [5,14,15,16]. Bioavailability is important as it describes the amount of a drug or nutrient absorbed into the systemic circulation in order to exert favourable bioactivity in vivo [1]. Bioaccessibility on the other hand denotes the amount of a compound accessible for absorption following digestion. Thus, analysing the bioaccessibility of polyphenols can help estimate their bioavailability [17]. For example, a feeding trial conducted by Gamel et al. [18] demonstrated that consumption of purple wheat resulted in a significant reduction in oxidative stress markers but a low bioavailability of anthocyanins. However, an investigation of the in vitro bioaccessibility would have elucidated the fate of the wheat anthocyanins following digestion, and thus determined the impact digestion had on the release of the anthocyanins. Interestingly, Tomé-Sánchez et al. [19] conducted in vitro gastrointestinal digestion of bioprocessed wheat products and identified the bioaccessible phenolic compounds to include ferulic derivatives and flavones. The study highlighted how different bioprocessing techniques influenced the bioaccessibility of polyphenols. This is in line with data showing how the food matrix and food processing have major influence on the bioaccessibility, bioavailability and subsequent health effects in vivo. [17].

In the human body, gastrointestinal digestion acts the physiological extractor of free phenolic compounds from cereals. However, because organic solvent extractions can obtain a better yield of phenolic content, they are a common method for characterisation of polyphenols [20,21]. Some studies on pigmented cereals have identified polyphenols and quantified their relative antioxidant levels the organic solvent extracts using an ultra-high performance liquid chromatography mass spectrometry coupled with an online 2,2′-azino-bis (3-ethylbenzothiazoline-6-sulfonic acid) (UHPLC-MS-Online ABTS^•+^) system [22,23]. Yet, a combination of extraction by organic solvents and simulated gastrointestinal digestion will identify antioxidant compounds and their subsequent bioaccessibility. Thus, the objectives of this study were (1) to compare the phenolic profile and antioxidant activity of three pigmented cereals: *Hordeum vulgare* (purple barley), *Triticum turgidum* (purple wheat) and *Triticum aestivum* (blue wheat), and (2) to determine the bioaccessibility and antioxidant activity of polyphenols in pigmented cereal after gastric and intestinal digestion.

## 2. Materials and Methods

### 2.1. Materials

Three genetically different pigmented hullless cereal grains were provided by Nisshin Flour Milling Co., Ltd. (Tokyo, Japan) and grown on the same field and under the same conditions in the same year (2021) in Narrabri, New South Wales by the Australian Grain Technologies. The pigmented cereal lines included *Hordeum vulgare* L. (Irisaka), *Triticum turgidum* (Triticum abyssinicum var arraseita) and *Triticum aestivum* (Sebesta Blue-3). These cereals were named purple barley, purple wheat and blue wheat, respectively based on the cereal type and the pigment colour.

Chemicals including methanol, hexane, acetonitrile, formic acid, acetic acid, sulphuric acid, anhydrous sodium acetate, ferric chloride, potassium chloride, sodium chloride, sodium bicarbonate and butylated hydroxytoluene (BHT) were acquired from Chem Supply Pty Ltd. (Port Adelaide, SA, Australia). Gallic acid, ferulic acid, catechin, luteolin, procyanidin B3, protocatechuic acid standards were purchased from Sigma-Aldrich (St. Louis, MO, USA). Digestive enzymes included bile extract, pancreatin and pepsin were sourced from Sigma-Aldrich (St. Louis, MO, USA). Hydrochloric acid (HCl), 6-Hydroxy-2,5,7,8-tetramethylchroman-2-carboxylic acid (trolox), 2,4,6-tris(2-pyridyl)-s-triazine (TPTZ), 2,2-diphenyl-1-picrylhydrazyl (DPPH), and 2,2′-azino-bis (3-ethylbenzothiazoline-6-sulfonic acid (ABTS) were sourced from Sigma-Aldrich (St. Louis, MO, USA).

### 2.2. Organic Solvent Extraction

The extraction was performed as described by Ed Nignpense et al. [21]. The pigmented cereal grains were ground using a Perten Laboratory Mill 3000 (Hägersten, Sweden) sieved through a 0.5 mm mesh membrane. The flour was defatted with hexane. Polyphenols were extracted from 10 g of defatted cereal flour using 100 mL methanol. The extracts were concentrated to 1 g mL^−1^ using a vacuum evaporator (Rotavapor R-210 BUCHI Labortechnik, Flawil, Switzerland).

### 2.3. Simulated In Vitro Gastrointestinal Digestion of Flour

The static in vitro digestion procedure described by Ed Nignpense et al. [21] was used to determine the bioaccessible amounts of polyphenols after the gastric and intestinal phases of digestion. Experiments were performed in triplicate to assess polyphenol bioaccessibility and antioxidant activity following both the gastric and intestinal phases. Flour was homogenised with saline solution (140 mM NaCl, 5 mM KCl, 150 µM BHT). Gastric digestion was conducted using 1 M HCl and pepsin solution (0.2 g of pepsin in 0.1 M HCl) and incubation at 37 °C in a water bath for 1 h followed by centrifugation at 4000 rpm for 10 min. Supernatants was retrieved for further analyses. Intestinal digestion was conducted after increasing pH of gastric digesta to 6.9 with sodium bicarbonate. Samples were mixed with pancreatic bile solution and incubated at 37 °C in a water bath for 2 h. The sample was centrifuged, and supernatants retrieved for analysis. Both gastric and intestinal phase supernatants were lyophilised and concentrated to 1 g mL^−1^ before analysis. A blank (reagents without the samples) was used to corroborate for interaction with enzymes and buffers utilised in the study.

### 2.4. Total Phenolic Content

The quantification of total free phenolic content was performed through Folin–Ciocalteu method described by Rao et al. [23]. The phenolic content was expressed milligrams of gallic acid equivalents (GAE) per 100 g of dry weight of cereal flour.

### 2.5. Ferric Reducing Antioxidant Power Assay

The ferric reducing antioxidant power (FRAP) method was adopted from Sompong et al. [24]. FRAP values were presented as milligrams of Trolox equivalents (TE) per 100 g dry weight of cereal flour.

### 2.6. DPPH Radical Scavenging Activity Assay

The method described by Sompong et al. [24] was used to evaluate the total DPPH free radical scavenging activity of the samples.

DPPH radical scavenging activity (%) was calculated as per the formula below:(1)$$\frac{\mathrm{Absorbance}\;\mathrm{of}\;\mathrm{control}-\mathrm{Absorbance}\;\mathrm{of}\;\mathrm{sample}\;}{\mathrm{Absorbance}\;\mathrm{of}\;\mathrm{control}}\times 100\%$$

### 2.7. ABTS^•+^ Radical Scavenging Activity Assay

The method described by Saji et al. [25] was used to determine the total ABTS^•+^ radical scavenging activity of the samples.

ABTS^•+^ radical scavenging activity (%) was calculated as per the formula below:(2)$$\frac{\mathrm{Absorbance}\;\left(\mathrm{control}\right)-\mathrm{Absorbance}\;\left(\mathrm{sample}\right)\;}{\mathrm{Absorbance}\;\left(\mathrm{control}\right)}\times 100\%$$

### 2.8. UHPLC Combined with Online ABTS^•+^

Chromatographic analysis to characterise cereal grain polyphenols and quantify antioxidant activity was carried out using an Agilent 1290 Infinity UHPLC system with a C18 Poroshell 120 column as described by Ed Nignpense et al. [21]. Polyphenol identification, anthocyanin identification and antioxidant activity quantification were achieved by a scanning DAD at wavelengths of 280 nm, 520 nm and 414 nm, respectively. Peaks were identified using retention time, peak spectra and the available standards. The detection of ABTS activity was recorded using a separate module coupled with UHPLC system having a binary pump injecting ABTS^•+^ working solution at a flow rate of 0.32 mL min^−1^, coil column and UV/Visible wavelength detector at 414 nm for polyphenols exhibiting antioxidant activity. The ABTS^•+^ working solution was prepared as described previously (Section 2.7). Compounds were quantified as gallic acid equivalents (GAE)/100 g dry weight of cereal flour. The ABTS^•+^ radical scavenging activity was expressed as Trolox equivalents (TE)/100 g dry weight of cereal flour.

### 2.9. Identification of Compounds

The UHPLC system coupled with an Agilent 6530 Accurate-Mass LC/MS Q-TOF (Agilent Technologies, Santa Clara, CA, USA) to generate mass spectra of molecular features. A complete mass spectral scan was collected in a range of 50–1300 *m/z*. The peaks were identified in negative mode acquired with capillary and nozzle voltage set at 3.5 kV and 500 V, respectively. Mass spectra was extracted using Agilent Mass Hunter Qualitative Analysis software version B07.00. Peaks were tentatively identified using published literature and databases such as Metlin (Agilent Technologies, Santa Clara, CA, USA) and online tools including ChemSpider, Pubchem and MassBank.

### 2.10. Statistical Analysis

The experiments were conducted in triplicates for extraction and analysis. All descriptive statistical analyses were performed using GraphPad Prism 9 software (GraphPad Software Inc., San Diego, CA, USA) and results are expressed as mean ± standard deviation where possible. A one-way analysis of variance (ANOVA) followed by Tukey’s post hoc multiple comparisons test was used to identify the significance of the difference between test groups. A *p*-value of less than 0.05 was considered as statistically significant.

## 3. Results

### 3.1. Total Phenolic Content of Extracts from Digested Cereal Flours Compared with Methanol Extracts

A significant difference (*p* < 0.001) in total phenolics was observed between methanol extracts of purple barley (78 ± 6.1 mg GAE/100 g dw), purple wheat (35 ± 0.4 mg GAE/100 g dw) and blue wheat (40 ± 2.1 mg GAE/100 g dw) (Figure 1). Using the quantities from methanol extracts as a baseline, the percentage of bioaccessible phenolics after each step of flour digestion was determined. Following digestion of purple barley flour, about 35% (28 ± 7.0 mg GAE/100 g dw) and 48% (38 ± 1.2 mg GAE/100 g dw) phenolics were bioaccessible in the gastric and intestinal phase, respectively. Following digestion of purple wheat flour, about 91% (32 ± 2.5 mg GAE/100 g dw) and 117% (41 ± 1.3 mg GAE/100 g dw) phenolics were bioaccessible in the gastric phase and the intestinal phase, respectively. Following digestion of blue wheat flour, about 65% (26 ± 6.6 mg GAE/100 g dw) and 90% (36 ± 4.0 mg GAE/100 g dw) phenolics were bioaccessible in the gastric and intestinal phase, respectively. The stepwise increase in the total bioaccessible phenolics from the gastric phase to intestinal phase across all the cereal varieties was significant. However, between cereal varieties, there were similar amounts of bioaccessible phenolics at the gastric and intestinal phases of digestion. Thus, there was no significant difference in the total bioaccessible phenolic content between cereal varieties.

### 3.2. Ferric Reducing Antioxidant Power of Extracts from Digested Cereal Flours Compared with Methanol Extracts

The pattern of antioxidant activity observed through FRAP assay was comparable to the pattern observed in the total phenolic content assay (Figure 1 and Figure 2). Purple barley methanol extracts showed a significantly higher antioxidant activity than purple wheat and blue wheat methanol extracts. Purple barley digestion resulted in 50 ± 1.8 mg TE/100 g dw and 60 ± 2.8 mg TE/100 g dw antioxidant activity in the gastric and intestinal phase, respectively. Purple wheat digestion resulted in 34 ± 0.5 mg TE/100 g dw and 54 ± 0.5 mg TE/100 g dw antioxidant activity in the gastric and intestinal phase, respectively. Blue wheat digestion resulted in 36 ± 1.0 mg TE/100 g dw and 46 ± 0.8 mg TE/100 g dw antioxidant activity in the gastric and intestinal phase, respectively. Across the cereal varieties there was a significant stepwise increase in antioxidant activity from gastric to intestinal phases of digestion. During the gastric phase, purple barley possessed the greatest FRAP values followed by the wheat varieties. No significant differences were observed between purple wheat and blue wheat in the gastric phase (*p* > 0.05). During the intestinal phase, purple barley possessed the most antioxidant activity followed by purple wheat and blue wheat (*p* < 0.001).

### 3.3. DPPH Radical Scavenging Activity of Extracts from Digested Cereal Flours Compared with Methanol Extracts

The overall trend of antioxidant activity from the DPPH radical scavenging assay was similar to the trend observed from the total phenolic content assay (Figure 1 and Figure 3). Purple barley methanol extracts (77%) possessed a significantly (*p* < 0.001) higher DPPH radical scavenging activity than purple wheat (54%) and blue wheat (53%) methanol extracts. There was no significant difference in DPPH radical scavenging activity between purple wheat and blue wheat methanol extracts (*p* > 0.5). During digestion of the cereals there was a significant progressive increase in DPPH radical scavenging activity from the gastric to intestinal phase (*p* < 0.001). Within cereal varieties, DPPH radical scavenging activity was at about 25% and 36% after gastric and intestinal digestion, respectively.

### 3.4. ABTS^•+^ Radical Scavenging Activity of Extracts from Digested Cereal Flours Compared with Methanol Extracts

The purple barley methanol extract (72%) had the highest significant ABTS^•+^ radical scavenging activity followed by blue wheat (39%) varieties and purple wheat (25%), respectively. This trend observed in the ABTS^•+^ was different to the trend observed in the other antioxidant assays (Figure 2, Figure 3 and Figure 4). During the gastric phase, purple barley (29%) possessed significantly greater ABTS^•+^ antioxidant activity than both wheat varieties. No significant difference was detected between purple wheat (17%) and blue wheat (13%) varieties after gastric digestion (*p* > 0.05). During the intestinal phase, purple barley (68%) had a similar ABTS^•+^ antioxidant activity to purple wheat (70%) but a significantly greater activity than blue wheat (57%); *p* < 0.01. Both pigmented wheat varieties had a greater ABTS^•+^ antioxidant activity after intestinal digestion than after methanol extraction (*p* < 0.001).

### 3.5. Phenolic Characterisation and Antioxidant Activity of Pigmented Barley and Wheat Methanol Extracts

A total of 33 compounds were identified in the UHPLC-MS profile of the pigmented cereal extracts (Table 1). These compounds were identified by either comparison spectra generated by using analytical standards or putatively characterized using several online databases and published literature. A comparative quantification of unidentified compounds can be found in the Appendix A.

The UHPLC-MS-Online ABTS^•+^ characterisation of the three pigmented cereal extracts showed a distinct phenolic composition and antioxidant activity (Table 1 and Table 2). Purple barley had the most polyphenols as well as corresponding antioxidant activity compared to the purple wheat and blue wheat. Chrysoeriol-7-O-glucuronide was the most abundant phenolic compound in purple barley at 2.7 ± 0.04 mg GAE/100 g dw followed by chrysoeriol at 1.37 ± 0.05 mg GAE/100 g dw, respectively. Apigenin-6-C-arabinoside-8-C-hexoside isomer 2 was the second most significant phenolic compound identified in purple wheat and blue wheat at 0.99 ± 0 mg GAE/100 g dw and 0.81 ± 0.01 GAE/100 g dw, respectively. Purple barley showed five antioxidant peaks whereas purple wheat and blue wheat showed only one antioxidant peak (Table 2, Figure 5, Figure 6 and Figure 7).

Across the varieties, prodelphinidin B3, quantified in purple barley, had the strongest antioxidant activity at 49.05 ± 0.00 mg TE/100 g dw (Table 2, Figure 5). Other free radical scavenging compounds within purple barley included procyanidin B3, catechin, unknown compound P3 and gallocatechin at 8.72 ± 2.10 mg TE/100 g dw, 8.26 ± 1.24 mg TE/100 g dw, 2.95 ± 0.35 mg TE/100 g dw and 1.88 ± 0.66 mg TE/100 g dw, respectively. The only active compound in scavenging free radicals within purple wheat and blue wheat was an unknown compound (P3) at 5.86 ± 1.04 mg TE/100 g dw and 2.52 ± 0.52 mg TE/100 g dw, respectively. The compound had a significantly greater antioxidant activity in purple wheat than in blue wheat (*p* < 0.05). Two tentatively identified anthocyanins, i.e., cyanidin 3-glucoside and malvidin 3-(6″-acetylglucoside) were detected in purple barley with only cyanidin 3-glucoside detected in trace amounts in purple wheat. No anthocyanins were detected in blue wheat (Table 1 and Appendix A). Anthocyanins detected did not have any quantifiable ABTS^•+^ antioxidant peak (Table 2). No quantifiable ABTS peaks were detected following digestion of the flours (Data not shown).

### 3.6. Bioaccessibility of Polyphenols during Simulated Digestion

As shown in Table 3, gastric digestion releases all the polyphenols detected from the methanol extraction of the cereals. However, there is a significant decrease in the total number of bioaccessible polyphenols from the gastric phase to the intestinal phase. The number of bioaccessible polyphenols decreased from 13 to 7 in purple barley. The number of bioaccessible polyphenols reduced from 7 to 4 and 7 to 5 for purple wheat and blue wheat, respectively. Chrysoeriol-7-O-glucuronide significantly increased in amounts from the gastric to intestinal phases of purple barley digestion. Examples of compounds with reduced bioaccessibility during the digestive process included flavan-3-ols (catechin and its derivates; gallocatechin, prodelphinidin B3, procyanidin B3), anthocyanins (cyanidin 3-glucoside and malvidin 3-(6″-acetylglucoside)) and apigenin-8-C-sinapoylpentoside-6-C-hexoside. Some polyphenols which retained similar levels in both phases of digestion included luteolin, apigenin-6-C-arabinoside-8-C-hexoside isomer 1 and apigenin-6-C-arabinoside-8-C-hexoside isomer 2.

## 4. Discussion

Polyphenols derived from pigmented cereals have been demonstrated to possess strong antioxidant properties that contribute to their numerous health benefits [18,28,32]. However, to exert these health effects, polyphenols are required to be bioavailable. Determining the bioaccessibility of polyphenols is a first step in the evaluation of their bioavailability. There is a scarcity of studies investigating the bioaccessibility of polyphenols in pigmented cereals [21,33,34]. In this study, a static model of upper gastrointestinal digestion was employed to evaluate polyphenol bioaccessibility. In conjunction with methanol extractions, the simulated digestion of the pigmented cereal flours highlighted differences in composition, bioaccessibility and antioxidant activity of polyphenols. It was hypothesised that the cereal with greater phenolic content would demonstrate greater phenolic bioaccessibility and antioxidant activity as bound phenolic compounds are released progressively during digestion. The results showed that the purple barley methanol extract had a higher total phenolic content and antioxidant activity in comparison with purple wheat and blue wheat extracts (Table 1, Figure 1, Figure 2, Figure 3 and Figure 4). Interestingly, after digestion of the flour, purple barley had a similar level of bioaccessible phenolic content to the wheat varieties but demonstrated the greatest antioxidant activity (Figure 1, Figure 2 and Figure 4).

Among methanol extracts, purple barley had the greatest total phenolic content and antioxidant activity in all benchtop assays (Figure 1, Figure 2, Figure 3 and Figure 4). This is likely due to the greater number of polyphenols in purple barley acting as stronger reducing agents and free radical scavengers (Figure 5). Results of benchtop antioxidant assays generally followed a similar trend (Figure 1, Figure 2, Figure 3 and Figure 4). Interestingly, blue wheat methanol extract had a greater ABTS^•+^ scavenging activity than purple wheat methanol extract although both wheat varieties had similar antioxidant activity from DPPH and FRAP assays (Figure 2 and Figure 3). This observation may potentially be due to specific antioxidant mechanisms that phenolic compounds from blue wheat synergistically target. FRAP assays measure the ferric reducing ability of compounds whereas both DPPH and ABTS^•+^ assays measure radical scavenging activity [24]. However, the ABTS^•+^ assay involves a single electron transfer reaction, while DPPH assay involves a transfer of hydrogen atom or electrons [7]. Interestingly, a study by Shen et al. [32] reported that black highland barley phenolic extract possessed strong free radical scavenging properties (67% DPPH antioxidant activity at 0.25 mg/mL) and could improve lipid profile, cellular antioxidant defence system and gene expression in mice consuming an extract at a dose of 600 mg/kg body weight. Considering the antioxidant potential of the phenolic extracts, future studies investigating their relative bioactivities using both in vitro and in vivo approaches are warranted.

Simulated digestion significantly impacted the bioaccessibility of antioxidant phenolic compounds, when measured using the benchtop assays (Figure 1, Figure 2, Figure 3 and Figure 4). Although purple barley methanol extract had a greater phenolic content, the bioaccessible phenolic content after simulated digestion was similar to that of the other cereals (Figure 1). This was likely due to the differences in the stability and solubility of polyphenols during the digestive process [16,35]. Some of the purple barley polyphenols, especially catechin and proanthocyanins, may have had lower solubility and stability in the digestive medium than in methanol (Table 1 and Table 3). As expected, there was an increasing trend of bioaccessible phenolic content from the gastric to intestinal phase as described in Figure 1. This can be explained by the enzymatic release of bound phenolic compounds during the digestive process [16]. Interestingly, the bioaccessible phenolics exhibited antioxidant activity progressively from gastric to intestinal phase (Figure 2, Figure 3 and Figure 4). Except for the DPPH assay, bioaccessible phenolics from purple barley had significantly greater FRAP values and ABTS^•+^ antioxidant activity than those of purple wheat and blue wheat after gastric digestion. This is likely due to the release of strong antioxidants; flavan-3-ols and anthocyanins in the gastric phase (Table 1). Following intestinal digestion, bioaccessible phenolics from purple barley exhibited ABTS^•+^ scavenging activity similar to those of purple wheat but greater than those of blue wheat. Purple barley bioaccessible phenolics also exhibited a stronger ferric reducing power than that of both wheat varieties (Figure 2 and Figure 4). Interestingly, purple wheat and blue wheat demonstrated a greater ABTS^•+^ antioxidant activity after intestinal digestion than after methanol extraction (Figure 4). This could be due to the release of specific bound phenolic and non-phenolic compounds that may possess free radical scavenging activity (Appendix A). Further research into identifying the other free radical scavenging compounds in the pigmented cereals is warranted.

Although benchtop antioxidant assays are a good screening tool for determining phenolic composition and antioxidant activity, they are prone to interferences by non-phenolic compounds [21]. Chromatographic techniques offer a more precise approach to identifying polyphenols. In this study, online ABTS^•+^ coupled with LC-MS was used to examine the phenolic composition of the cereals, and to identify the polyphenols that govern the free radical scavenging activity (Table 1 and Table 2). Phenolic characterisation of the methanol extract identified bioactive polyphenols falling into the class of flavone glycosides, phenolic acids, anthocyanins, and flavan-3-ols (Table 1). Surprisingly, anthocyanins were not the dominant antioxidant compounds across the pigmented cereals (Table 2). The levels of anthocyanins extracted from the pigmented wheat varieties were lower compared to previously published literature [28,36]. This could be due to different extraction protocols used in other studies. Other researchers have employed extraction protocols that involved longer extraction times, stages of acidification and organic solvent extraction [37,38,39]. These protocols may have allowed for a more efficient extraction of polyphenols bound to arabinoxylans or cell wall polysaccharides. Interestingly, the present study demonstrated that the anthocyanins were released after gastric digestion but not after intestinal digestion. Although this reduced intestinal bioaccessibility is likely to be due to instability in the alkaline medium, the detection in the gastric phase is noteworthy. Anthocyanins may be absorbed in the stomach to exert their health-promoting effects [4,23,40].

The compounds with the dominant antioxidant activity included an unknown compound and flava-3-nols extracted in wheat and barley, respectively (Table 1 and Table 2). Flavan-3-ols have been previously shown to be the major compound responsible for free radical scavenging activity in barley [22,23,27]. However, to the best our knowledge this is the first study to evaluate the online ABTS^•+^ antioxidant activity of wheat extracts. The dominant antioxidant compound P3 in pigmented wheat is likely to contribute its antioxidant activity and subsequent health promoting properties—thus warranting further investigation. Furthermore, to the best of our knowledge this the first study to evaluate the bioaccessibility of barley flavan-3-ols following digestion. The current study indicated that barley flavan-3-ols were extracted in the gastric phase but not bioaccessible after intestinal digestion likely due to instability and degradation in alkaline medium (Table 3). Nevertheless, even with their low bioaccessibility, it has been suggested that flavan-3-ols may form complexes with protein and act as radical scavengers in the lumen—thus promoting gastrointestinal health [38,41,42].

Some polyphenols were bioaccessible after intestinal digestion. Flavones and flavone glycosides including luteolin, chrysoeriol and apigenin-6-C-arabinoside-8-C-hexoside isomers were bioaccessible during the digestive process. This is consistent with results from the study by Tomé-Sánchez et al. [19] which showed that flavones were stable during simulated digestion of bioprocessed wheat. The flavone chrysoeriol-7-O-glucuronide in purple barley showed the most bioaccessibility with a significant stepwise release during simulated digestion (Table 3). These findings are noteworthy as flavones and flavone-rich cereals have been reported to possess strong anti-inflammatory activity [23,41]. In contrary to other studies, ferulic acid, a phenolic acid common in cereals, was not quantified in this study [37,38,39]. Rather, it was identified by LC-MS analysis (Appendix A). This may be due to different extraction protocols used in other studies as described earlier with regard to anthocyanin extraction [37,38,39]. Future studies may benefit in the use of these extraction protocols where the quantification of ferulic acid and similar monomeric phenolic acids are of interest. The phenolic acid protocatechuic acid was identified in all cereal varieties—being most bioaccessible from the digestion of purple barley (Table 3). Unidentified compounds detected in pigmented cereals likely contributed to the measured bioaccessible phenolic content (Table 1 and Appendix A). Hence, future research should employ a more comprehensive, and detailed MS/MS data analysis for identification of unknowns.

## 5. Conclusions

The present study demonstrated that purple barley, purple wheat and blue wheat are sources of bioaccessible antioxidant phenolic compounds. The nutraceutical value of the pigmented cereals was highlighted by their benchtop analysis results, wherein the barley methanol extract had a greater total phenolic content and antioxidant activity than the wheat varieties. However, after digestion of the flour there was no significant difference in the bioaccessible phenolic content between the cereal varieties. Overall, the results suggested that the gastrointestinal tract may act as an extractor where polyphenols are released progressively from the cereal matrix and made available for either bioactivity in the gastrointestinal tract or intestinal absorption. Individual compounds were shown to behave differently during the transition from gastric to intestinal digestion—with flavones being more bioaccessible than anthocyanins and flavan-3-ols. The findings of this study could aid in the selection of appropriate processing techniques that would be instrumental in the design of novel functional foods with nutraceutical benefits.

## Figures and Tables

**Figure 1 foods-11-03697-f001:**
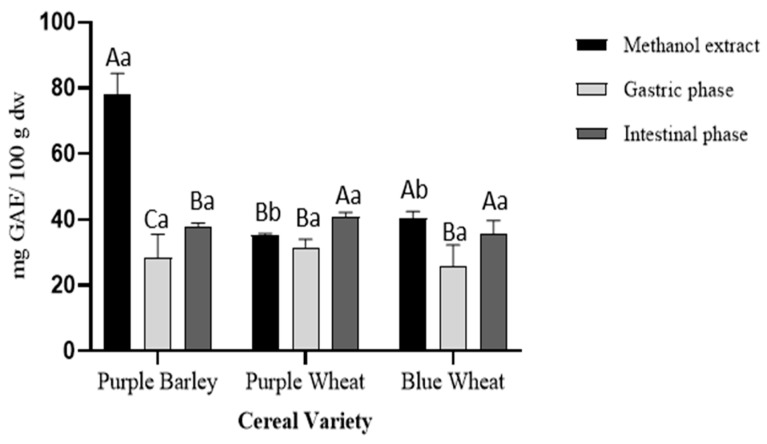
Total phenolic content of extracts from digestion of pigmented cereal flour compared with a methanol extract using a Folin–Ciocalteu method. Data shown is blank corrected, expressed as mg GAE/100 g dry weight cereal flour and represent mean ± SD; *n* = 3. Different capitalized alphabets represent significant differences within a cereal variety. Different lowercase alphabets represent significant differences between cereal varieties.

**Figure 2 foods-11-03697-f002:**
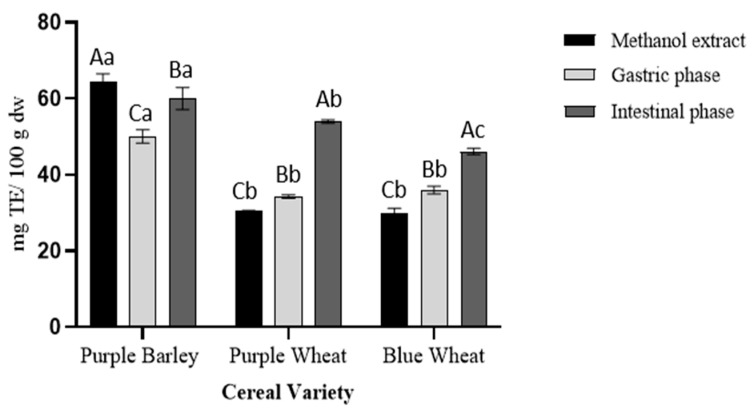
Total antioxidant activity of extracts from digestion of pigmented cereal flour compared with a methanol extract using a FRAP assay. Data shown is blank corrected, expressed as mg TE/100 g dry weight cereal flour and represent mean ± SD; *n* = 3. Different capitalized alphabets represent significant differences within a cereal variety. Different lowercase alphabets represent significant differences between cereal varieties.

**Figure 3 foods-11-03697-f003:**
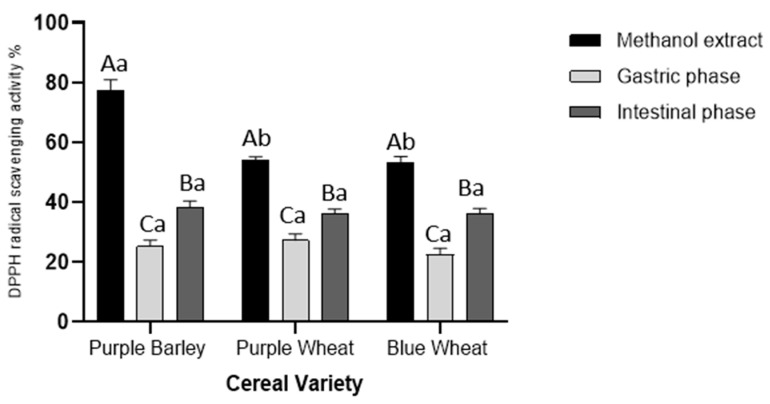
Total antioxidant activity of extracts from digestion of pigmented cereal flour compared with a methanol extract measured using the DPPH radical scavenging assay. Data shown is blank corrected, expressed as percentages and represents mean ± SD; *n* = 3. Different capitalized alphabets represent significant differences within a cereal variety. Different lowercase alphabets represent significant differences between cereal varieties.

**Figure 4 foods-11-03697-f004:**
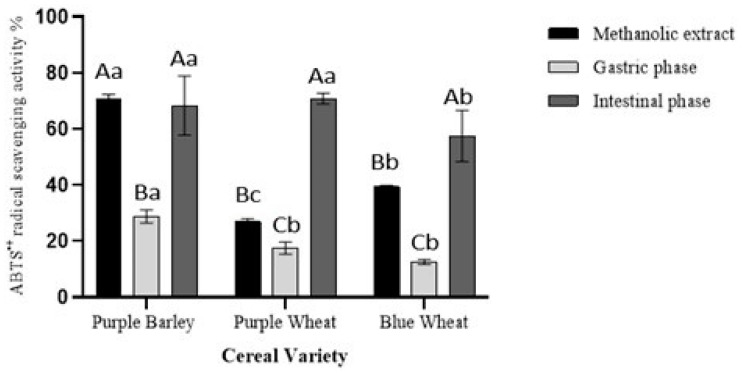
Total antioxidant activity extracts from digestion of pigmented cereal flour compared with a methanol extract measured using the ABTS^•+^ radical scavenging assay. Data shown is blank corrected, expressed as percentages and represents mean ± SD; *n* = 3. Different capitalized alphabets represent significant differences within a cereal variety. Different lowercase alphabets represent significant differences between cereal varieties.

**Figure 5 foods-11-03697-f005:**
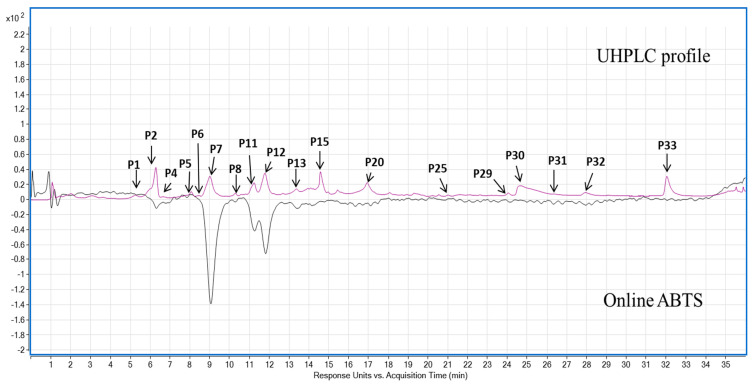
Ultra-high performance liquid chromatography (UHPLC) coupled with an online 2,2′-azino-bis (3-ethylbenzothiazoline-6-sulfonic acid) (ABTS^•+^) antioxidant activity mapping of polyphenols from purple barley methanol extract. Purple line: phenolic profile at 280 nm; Black line: online ABTS antioxidant activity profile at 414 nm.

**Figure 6 foods-11-03697-f006:**
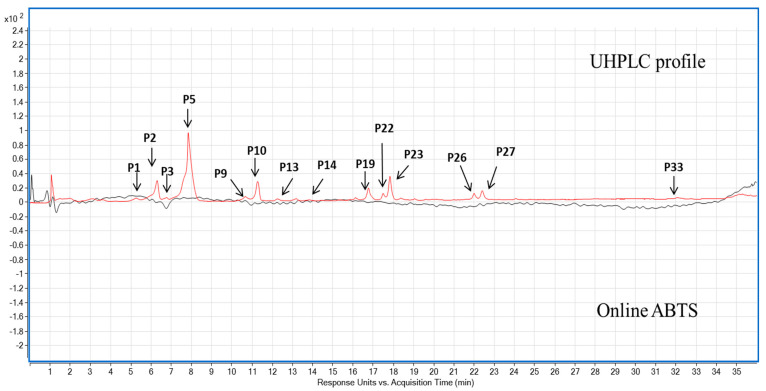
Ultra-high performance liquid chromatography (UHPLC) coupled with an online 2,2′-azino-bis (3-ethylbenzothiazoline-6-sulfonic acid) (ABTS^•+^) antioxidant activity mapping of polyphenols from purple wheat methanol extract. Red line: phenolic profile at 280 nm; Black line: online ABTS antioxidant activity profile at 414 nm.

**Figure 7 foods-11-03697-f007:**
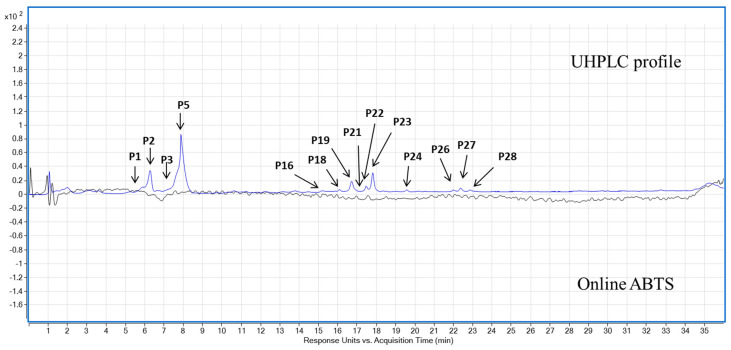
Ultra-high performance liquid chromatography (UHPLC) coupled with an online 2,2′-azino-bis (3-ethylbenzothiazoline-6-sulfonic acid) (ABTS^•+^) antioxidant activity mapping of polyphenols from blue wheat methanol extract. Blue line: phenolic profile at 280 nm; Black line: Online ABTS antioxidant profile at 414 nm.

**Table 1 foods-11-03697-t001:** Quantification (mg GAE/100 g dw) of barley and wheat polyphenols at 280 nm.

Peak	RT (min)	Λmax (nm)	*m/z*	Tentative Identification	Phenolic Quantification (mg GAE/100 g dw)			Class	Reference
					Purple Barley	Purple Wheat	Blue Wheat		
P1	5.7	280	153.0208	Protocatechuic acid	0.06 ± 0.02 ^a^	0.10 ± 0.04 ^a^	0.10 ± 0.03 ^a^	Phenolic acid	Standard
P4	7.2	280	305.0667	Gallocatechin	0.07 ± 0.01	–	–	Flavan-3-ol	[26]
P7	9.5	280	593.1287	Prodelphinidin B3	0.91 ± 0.13	–	–	Flavan-3-ol	[27]
P11	11.8	280	289.0723	Catechin	0.58 ± 0.17	–	–	Flavan-3-ol	Standard
P12	11.9	280	577.1345	Procyanidin B3	0.75 ± 0.16	–	–	Flavan-3-ol	Standard; [27]
P13	13.6	280, 520	447.0925	Cyanidin 3-glucoside	0.7 ± 0.15	trace	–	Anthocyanin	[6,28]
P19	17.1	270, 340	563.1422	Apigenin 6-C-arabinoside-8-C-hexoside	–	0.49 ± 0 ^a^	0.50 ± 0.01 ^a^	Flavone glycoside	[29]
P20	17.5	320, 520	533.0932	Malvidin 3-(6″-acetylglucoside)	1.06 ± 0.07	–	–	Anthocyanin	[6,28]
P22	17.8	270, 240	563.1422	Apigenin-6-C-arabinoside-8-C-hexoside isomer 1	–	0.22 ± 0 ^a^	0.22 ± 0 ^a^	Flavone glycoside	[29]
P23	18.2	270, 340	563.1422	Apigenin-6-C-arabinoside-8-C-hexoside isomer 2	–	0.99 ± 0 ^a^	0.81 ± 0.01 ^b^	Flavone glycoside	[29]
P25	21.5	270, 350	461.1077	Chrysoeriol-7-O-glucoside	0.39 ± 0.03	–	–	Flavone glycoside	[30]
P26	22.4	330	769.1973	Apigenin-8-C-sinapoylpentoside-6-C-hexoside	–	0.27 ± 0 ^a^	0.1 ± 0 ^b^	Flavone glycoside	[5]
P27	22.8	330	769.1973	Apigenin-8-C-sinapoylpentoside-6-C-hexoside isomer	–	0.43 ± 0 ^a^	0.18 ± 0 ^b^	Flavone glycoside	[5]
P28	23.2	330	739.1866	Chrysoeriol 4′-O-pentoside-7′ O-rutinoside	–	–	0.07 ± 0	Flavone glycoside	[31]
P29	24.4	270, 350	461.1077	Chrysoeriol-7-O-glucoside isomer 1	0.24 ± 0	–	–	Flavone glycoside	[30]
P30	24.7	270, 350	475.0874	Chrysoeriol-7-O-glucuronide	2.7 ± 0.04	–	–	Flavone glycoside	[30]
P31	24.9	350	461.1077	Chrysoeriol-7-O-glucoside isomer 2	0.09 ± 0	–	–	Flavone glycoside	[30]
P32	28.2	270, 350	285.0391	Luteolin	0.34 ± 0.01	–	–	Flavone	Standard
P33	32.5	270, 350	299.0558	Chrysoeriol	1.37 ± 0.05 ^a^	0.08 ± 0 ^b^	–	Flavone	[14]

Data are the means ± SD (*n* = 3). Different alphabets in each row indicates a significant difference in phenolic content. Gallic acid equivalent: GAE; Mass to charge ration: *m/z*; – not detected; RT: retention time; trace: compound below limit of quantification.

**Table 2 foods-11-03697-t002:** Comparison of Online ABTS^•+^ antioxidant activity (414 nm) in pigmented barley and wheat methanol extracts.

Peak	Tentative Identification	ABTS^•+^ Antioxidant Activity (mg TE/100 g dw)		
		Purple Barley	Purple Wheat	Blue Wheat
P3	Unknown	–	5.86 ± 1.04 ^a^	2.52 ± 0.52 ^b^
P4	Gallocatechin	4.94 ± 1.01	–	–
P7	Prodelphinidin B3	49.05 ± 0.00	–	–
P11	Catechin	8.72 ± 2.10	–	–
P12	Procyanidin B3	8.26 ± 1.24	–	–

Data are the means ± SD (*n* = 3). Different alphabets in each row indicated a significant difference in antioxidant activity. – not detected. TE: Trolox equivalent.

**Table 3 foods-11-03697-t003:** Quantification of polyphenols measured at 280 nm using UHPLC from post digestion at gastric and intestinal phases.

Peak	Tentative Identification	Purple Barley mg GAE/100 g dw		Purple Wheat mg GAE/100 g dw		Blue Wheat mg GAE/100 g dw	
		Gastric	Intestinal	Gastric	Intestinal	Gastric	Intestinal
P1	Protocatechuic acid	1.22 ± 0.66 ^a^	0.92 ± 0.45 ^a^	0.691 ± 0.164	–	0.469 ± 0.104	–
P4	Gallocatechin	0.13 ± 0.07	–	–	–	–	–
P7	Prodelphinidin B3	0.188 ± 0.00	–	–	–	–	–
P11	Catechin	0.196 ± 0.07	–	–	–	–	–
P12	Procyanidin B3	0.192 ± 0.04	–	–	–	–	–
P13	Cyanidin 3-glucoside	0.158 ± 0.01	–	–	–	–	–
P19	Apigenin 6-C-arabinoside-8-C-hexoside	–	–	0.529 ± 0.05 ^b^	0.805 ± 0.09 ^a^	0.905 ± 0.25	trace
P20	Malvidin 3-(6″-acetylglucoside)	–	–	–	–	–	–
P22	Apigenin-6-C-arabinoside-8-C-hexoside isomer 1	–	–	0.687 ± 0.27 ^a^	0.877 ± 0.15 ^a^	0.409 ± 0.17 ^a^	0.470 ± 0.07 ^a^
P23	Apigenin-6-C-arabinoside-8-C-hexoside isomer 2	–	–	1.50 ± 0.17 ^a^	1.67 ± 0.15 ^a^	1.975 ± 0.29 ^a^	1.59 ± 0.30 ^a^
P25	Chrysoeriol-7-O-glucoside	0.47 ± 0.30 ^a^	0.72 ± 0.17 ^a^	–	–	–	–
P26	Apigenin-8-C-sinapoylpentoside-6-C-hexoside	–	–	1.26 ± 0.32 ^a^	0.483 ± 0.04 ^b^	0.903 ± 0.11	trace
P27	Apigenin-8-C-sinapoylpentoside-6-C-hexoside isomer	–	–	0.703 ± 0.09	–	trace	–
P28	Chrysieriol 4′-O-pentoside-7′ O-rutinoside	–	–	–	–	trace	trace
P29	Chrysoeriol-7-O-glucoside isomer 1	0.26 ± 0.09 ^b^	0.56 ± 0.12 ^a^	–	–	–	–
P30	Chrysoeriol-7-O-glucuronide	4.64 ± 1.81 ^b^	9.93 ± 1.74 ^a^	–	–	–	–
P31	Chrysoeriol-7-O-glucoside isomer 2	0.30 ± 0.18	trace	–	–	–	–
P32	Luteolin	0.33 ± 0.16 ^a^	0.30 ± 0.03 ^a^	–	–	–	–
P33	Chrysoeriol	0.36 ± 0.09 ^b^	0.58 ± 0.10 ^a^	trace	–	–	–
	No. of compounds detected	13	7	7	4	7	5

Data presented here are the means ± SD (*n* = 3 independent experiments), the different letters within the same row represent significant differences between gastric and intestinal at the 0.05 level. – not detected; trace: compound below limit of quantification.

## Data Availability

Data is contained within the article or Appendix A.

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
