# Peer review of "Bioaccessibility and Antioxidant Activity of Polyphenols from Pigmented Barley and Wheat"

_foods, 2022, doi:10.3390/foods11223697_

Round 1

Reviewer 1 Report

·         The abstract explains the purpose of the work and includes the background information

·         The introduction provides a good general background on the subject and gives the reader an idea of the wide range of possible applications of this technology.

·         The methods used in this thesis are appropriate for the goal of the study.

However, the manuscript has very questionable statistical analysis results. Looking at the results presented, major illogicalities become apparent. We suggest that the authors repeat the statistical results and then completely revise the manuscript.

Author Response

Thank you for taking the time to review the manuscript and provide comments and suggestions. Your feedback was very valuable in making the necessary amendments.

Reviewer 2 Report

General           

The manuscript discussed polyphenols' bioaccessibility and antioxidant activity from 3 different cereal grains without any treatment. The authors have written the manuscript in coherent and consistent English. However, there are some issues that need to be improved. The study is merely like an analysis of ingredients for the primary research regarding using these cereals for other purposes or processes. Please clarify why this report is essential and in line with the scoop of Foods journal. There is a page numbering issue in that pages are not numbered correctly.

Introduction

The description of three different pigmented cereals is not described clearly in the introduction. There is also no adequate explanation for why the authors used these cereals for this study. Please improve the introduction to strengthen the scientific background.

Discussion

Page 3 line 95: “…..previous published methods from our lab by…” Please improve this sentence as it shows the conflict of interest using a reference from a colleague.

Page 3 line 98:  “…defatted cereal flour…” Was the flour being defatted? If there were any, please describe the methods.

Page 3 line 106:  “Independent experiments were performed to assess polyphenol bioaccessibility and antioxidant activity following both the gastric and intestinal phases.” Please clarify why the study used different samples for gastric and intestinal phases as this may result in more error and less accuracy.

Page 3 line 100: “Three independent experiments were conducted for extraction and analysis.” Please move this sentence to the Statistical analysis section for all analyses (otherwise stated in the section) as other sections of methods do not have this explanation.

Author Response

(The authors gave the same response as above.)

Reviewer 3 Report

The manuscript “Bioaccessibility and antioxidant activity of polyphenols from pigmented barley and wheat” deals with the bioaccessibility study of purple barley, purple wheat and blue wheat. The polyphenols and antioxidant activities were determined, which showed the relevance in the development of functional food. The manuscript is well-written and compiled. However, there are some concerns which need to be addressed before the final decision is made.

Comments

·       In the abstract section, some future thrust line can be incorporated.

·       LN 38-40: The authors must include some other chemical properties of polyphenols.

·       LN 48-50: Please rewrite the line.

·       LN 55: Please provide a reference here.

·       LN 58: Few facts on the recent updates on bioaccessibility of phytonutrients can be included here.

https://doi.org/10.1016/j.tifs.2020.01.019

·       The objectives of the study is well defined.

·       LN 93: Please check for the formatting of the line.

·       LN 119: Please write the method used, not the chemical used. Eg: total free phenolic content

·       Similarly LN 124.

·       LN 197: Please write the interpretation from the result.

·       Kindly rewrite the subheadings of all the results.

·       Ln 258-260: Please rewrite the line.

·       The discussion section is well-represented. However, kindly improve the language to some extent.

·       The conclusion section needs to be improvised properly.

Author Response

(The authors gave the same response as above.)

Round 2

Reviewer 1 Report

Dear authors,

Thank you for the replies, but I am not sure if you understood what I specifically meant when I commented on the statistical processing of the data, i.e., the display of the results. I will try to explain as concretely as possible what you are doing wrong when you display the statistically processed results.

Figure 1: Please check the letters of the ANOVA /Tukey test in the graph. It appears that the letters are not correctly assigned to the mean values. For example, the methanol extract of PB (Purple Barley) is "~78A", the gastric phase value is "~30B", and the intestinal phase value is "~38C", which is incorrect. The letters should denote the mean values in order from low to high (or vice versa). The same error is seen in the other columns/figures. I suggest that the authors seek the help of statisticians, who will greatly improve the scientific soundness of this paper by applying statistical methods correctly.

Author Response

Thank you again for taking the time to review the manuscript and provide comments and suggestions. We have made amendments as per your comments.

Reviewer 2 Report

The authors have answered all questions and improved the manuscript.

Author Response

Thank you for taking the time to review the manuscript. Your feedback was very valuable in making the necessary amendments.

Reviewer 3 Report

The author made significant changes in the manuscript. I congratulate author for nice work.

Author Response

(The authors gave the same response as above.)
